# Evaluating the Adoption of Evidence-Based Management Practices in Eye Hospitals

**DOI:** 10.3390/healthcare13030222

**Published:** 2025-01-22

**Authors:** Ganesh-Babu B. Subburaman, Sachin Gupta, Thulasiraj Ravilla, Helen Mertens, Carroll A.B. Webers, Frank J.H.M. van den Biggelaar, Maaike van Zuilen, Balagiri Sundar, Frits van Merode

**Affiliations:** 1Care and Public Health Research Institute (CAPHRI), Maastricht University, P.O. Box 616, 6200 MD Maastricht, The Netherlands; f.vanmerode@maastrichtuniversity.nl; 2LAICO, Aravind Eye Care System, Madurai 625020, India; thulsi@aravind.org (T.R.); balagiri@aravind.org (B.S.); 3SC Johnson College of Business, Cornell University, Ithaca, NY 14853, USA; sachin.gupta@cornell.edu; 4Executive Board, Maastricht University Medical Centre+, P. Debyelaan 25, 6229 HX Maastricht, The Netherlands; helen.mertens@mumc.nl; 5Maastricht University Medical Center+, University Eye Clinic Maastricht, P.O. Box 5800, 6202 AZ Maastricht, The Netherlands; c.webers@mumc.nl (C.A.B.W.); f.vanden.biggelaar@mumc.nl (F.J.H.M.v.d.B.); 6World Association of Eye Hospitals, Schiedamse Vest 180, 3011 BH, Rotterdam, The Netherlands; maaike.vanzuilen@waeh.org; 7Maastricht University Medical Centre+, P. Debyelaan 25, 6229 HX Maastricht, The Netherlands

**Keywords:** internal data, internal expertise, evidence-based management

## Abstract

**Background:** Delivering sustainable, high-quality eye care requires a comprehensive understanding of patient conditions, clinical evidence, patients’ preferences, demand patterns, quality supplies, outcomes, financial sustainability, and satisfaction metrics. Evidence-based management (EBM) offers a structured approach to align actions with evidence, enabling effective decision-making and better organizational outcomes. Evaluating current practices against EBM principles fosters awareness and promotes an EBM culture in eye hospitals, supporting improved and sustainable service delivery. **Methods:** A descriptive cross-sectional survey was conducted in 2023–2024 among 94 eye hospitals worldwide, selected from two networks, using the Centre for Evidence-Based Management (CEBM) assessment questionnaire. Follow-up reminders resulted in 43 responses. Data analysis utilized frequency distributions and Pearson’s correlation to explore relationships between variables. **Results:** A strong positive correlation was observed between data accessibility and the capacity to interpret data in fostering evidence-based decision-making (r = 0.69, *p* < 0.01). Additionally, leveraging internal expertise and engaging stakeholders in assessing and utilizing data showed a moderate association with EBM practices (r = 0.48, *p* < 0.01). **Conclusions:** Eye hospitals demonstrate alignment with EBM principles, though regional variations exist. Organizations with robust data utilization systems, analytical expertise, and a commitment to continuous improvement are more effective in practicing EBM. Educational and peer-learning initiatives can further support hospitals in adopting EBM principles, strengthening their capacity for evidence-based decision-making, and enhancing eye care services.

## 1. Introduction

In a hospital setup, clinicians primarily focus on patients’ health, striving to deliver the best possible care to achieve effective clinical outcomes. Their decisions are often driven by patient-specific data, clinical evidence, and real-time observations, ensuring personalized and timely interventions [1,2]. On the other hand, healthcare managers are responsible for the organization’s overall health. Healthcare administrators are organizational leaders whose decisions have significant impacts on the effectiveness of delivering quality patient care and on the success of healthcare organizations. Key decisions that affect patient safety and quality of care and access, as well as widespread demands for reducing the cost of care and value-based purchasing, require healthcare administrators to take an evidence-based approach [3,4]. Because they are accountable for both their patients and their organizations, it is important for healthcare administrators to adopt an evidence-based management (EBM) approach in decision-making [5].

Both clinicians and managers need access to relevant, reliable and timely data to diagnose a problem and to determine the best possible solutions. Extensive research has been conducted to assess the barriers and enablers for EBM among healthcare managers. Systematic reviews of the EBM literature have identified external factors (1), contextual factors (2), resource availability (3), policies and procedures (4), research capacity (5), and data availability (6) as critical influences on EBM adoption [6]. Similar findings have been corroborated by other studies [7,8,9].

Liang et al. (2011) [10] assessed seven types of evidence commonly used by health managers: (1) internally developed data, (2) best practices, (3) stakeholder/consumer preferences, (4) examples of external practice, (5) expert opinions, and (6) quantitative and (7) qualitative research. Health managers rated internally developed data—information developed within the organization—as the most important and most frequently used [10]. Internal data reflect the specific dynamics, challenges, and outcomes within an organization, making internal data a preferred resource for informed decision-making. The quality, accuracy, and timeliness are fully within the organization’s control, emphasizing the need to equip managers with tools and systems to effectively access and utilize the data to drive performance improvements.

Developments in information and communication technology have greatly improved data availability and accessibility, allowing healthcare professionals to effectively utilize real-time data [11]. Electronic Health Records and Clinical Decision Support Systems enable clinicians to access patient histories instantly, recommend diagnostics, monitor outcomes, and provide comparative case details, thus enhancing clinical decision-making. Artificial-intelligence-based systems help to diagnose conditions, suggest the most appropriate interventions and also predict the outcome of the treatment [12,13]. Similarly, healthcare managers utilize real-time data analytics to optimize operations, track departmental performance, and align processes with organizational goals. These technologies bridge gaps in information flow, enhancing decision-making and resource efficiency across the healthcare continuum [14,15].

Despite this progress, evidence-based management in healthcare varies widely across regions and organizational types due to factors such as organizational culture, the existence of a practicable systematic process for improvements, legal requirements, technological advancements, and institutional priorities [16,17,18,19]. Currently, organizational culture is a significant obstacle to data-driven healthcare [20]. The value of benchmarking as a tool for driving improvements in the adoption of EBM has consistently been demonstrated, and is particularly the case when organizations openly share contextual details and specific activities underlying their practices [21,22].

This study explores the adoption of evidence-based management (EBM) practices among eye hospitals affiliated with the World Association of Eye Hospitals (WAEH) and partner hospitals of the Lions Aravind Institute of Community Ophthalmology (LAICO), with a specific focus on data-driven decision-making processes. By assessing the current level of adoption and identifying opportunities for improvement, the research aims to enhance the implementation of EBM practices across these institutions. The study seeks to foster cross-learning opportunities that encourage the proactive adoption of data-driven approaches to optimize clinical outcomes, operational efficiency, and strategic planning. Ultimately, this research contributes to strengthening data utilization and evidence-based approaches across the global eye care community.

The remainder of this manuscript is organized as follows: The subsequent sections outline the survey administration process, questionnaire validation, response grouping and scoring, geographic distribution of participants, alignment of current practices with evidence-based management (EBM) principles, and thematic focuses of management practices across regions. Furthermore, these sections examine the relationships between management practices and identify practices that hinder alignment with EBM principles, and they conclude with a comprehensive discussion of the findings.

## 2. Methodology

We conducted a descriptive cross-sectional survey aimed at evaluating evidence-based management practices among eye hospitals.

### 2.1. Participants

The study targeted eye hospitals globally, selecting partner hospitals of the Lions Aravind Institute of Community Ophthalmology (LAICO), Madurai, India, and member eye hospitals of the World Association of Eye Hospitals (WAEH).

LAICO (https://laico.org, accessed on 12 December 2024), the training and consulting division of Aravind Eye Care System in India, was founded in 1992 with a mandate to support eye care programs globally.

WAEH (https://www.waeh.org, accessed on 12 December 2024), a global network of independent specialized eye hospitals, promotes collaborative learning and the exchange of best practices.

### 2.2. Sampling Method and Sample Size

We invited all 61 member eye hospitals of the World Association of Eye Hospitals (WAEH) and employed a convenient sampling approach to select and invite 33 partner eye hospitals associated with the LAICO active collaborative network. The selection criteria focused on eye hospitals that had been actively engaged during the three years preceding the survey.

### 2.3. Survey Questionnaire

We utilized the evidence-based management (EBM) assessment questionnaire [23] developed by the Centre for Evidence-Based Management (CEBMa) to evaluate evidence-based practices within an organization. As noted on the CEBMa website, the tool has not been formally validated.

CEBMa (https://cebma.org, accessed on 12 December 2024), is a leading non-profit organization dedicated to advancing evidence-based practices in management and leadership. Established by an international network of scholars and practitioners, the centre focuses on promoting evidence-based decision-making by offering training, tools, and guidance to managers, educators, consultants, and other professionals. CEBMa’s work is geared towards fostering decision-making processes that are both data-driven and contextually relevant.

We used the same questionnaire in its English version for all respondents. Considering the profile of the respondents, translations were deemed unnecessary.

### 2.4. Data Collection

The survey questionnaire was administered using Google Forms, a publicly available tool, and included 19 items, each employing a 7-point scale with response categories ranging from “Never” to “Always” and “None of them” to “All of them”. Participant responses were automatically recorded and stored in Google Sheets, facilitating efficient data collection and organization for analysis. Respondents’ identification-based details, including both institutional and individual names, were securely anonymized and stored to ensure confidentiality.

### 2.5. Validation and Reliability of the Questionnaire

The study team conducted a thorough review to assess the alignment of the questionnaire with the study’s objectives, the clarity and comprehensibility of the questions, and the logical order of their presentation. Based on this review, minor modifications were made, including the addition of clarifying terms to enhance informativeness. Furthermore, an additional question was included to determine whether hospitals followed a systematic improvement process for addressing problems, bringing the total number of questions to 19. These refinements were implemented to enhance the questionnaire’s relevance, reliability, and suitability for the study’s context.

The survey was initially tested with an internal group of seven managers at Aravind Eye Hospital, Madurai. These participants completed the survey and were subsequently interviewed to provide feedback on the clarity of the questions and the ease of selecting appropriate responses. While no significant concerns were identified, two questions were reordered in the final questionnaire (Appendix A: Questionnaire) to enhance navigation and improve survey flow. These adjustments were finalized before administering the survey to the study hospitals. Participant responses were used to validate the internal consistency of the survey instrument, with the determined Cronbach’s alpha reliability coefficient of 0.84 indicating strong reliability.

### 2.6. Survey Administration

The survey was conducted in two phases: the first phase ran from October 2023 to December 2023 and targeted partner hospitals of LAICO, which were primarily located in Asia. The second phase took place from April 2024 to June 2024 and focused on members of WAEH, encompassing regions worldwide, excluding Africa. This phased approach facilitated tailored communication and systematic follow-up with participants, significantly improving response rates and ensuring the overall quality and reliability of the data collected.

#### 2.6.1. Survey of LAICO Cohort of Hospitals

The survey was sent to selected partner hospitals of LAICO, which had previously undergone capacity-building training at different times. A covering letter explaining the purpose of the survey, along with details of who should respond, was sent via email to the heads of the organizations. Participant responses were evaluated for data reliability using Cronbach’s alpha, which yielded a value of 0.86, indicating strong internal consistency.

#### 2.6.2. Survey of WAEH Cohort of Hospitals

The survey was distributed via email to all member hospitals by the global lead of WAEH, accompanied by a covering letter explaining the purpose of the survey, requesting heads of the organizations to complete it, and providing contact details for queries. Additionally, the survey details were included in the association’s monthly newsletter to enhance visibility and encourage participation. To ensure higher response rates, reminder emails were sent twice to members who had not yet completed the survey.

### 2.7. Classification of Questions

All survey questions were categorized into the following four groups based on their thematic focus (Annexure):I.External References and Guidance: This group includes four questions designed to assess the extent to which organizational decisions are influenced by external inputs. These influences include practices such as fostering innovation through external insights, benchmarking to identify and adopt best practices from other organizations, emulating strategies followed by peer institutions, and consulting external experts or advisors.II.Internal Expertise and Knowledge: Comprising five questions, this category evaluates the involvement of internal experts and stakeholders in the decision-making process, reflecting the organization’s reliance on in-house knowledge.III.Data Accessibility and Comprehension Skills: Focused on the organization’s ability to handle data effectively, this category includes four questions relating to accessing, understanding, and using internal data for decision-making.IV.Evidence-Based Practices: The largest category, consisting of six questions, examines the presence of processes and practices that promote evidence-based management within the organization.

This categorization ensures a structured approach to collecting and analysing the responses, enabling targeted insights into various facets of evidence-based management practices.

### 2.8. Scoring of Responses

We followed the scoring methodology provided in the CEBMa assessment tool [23]. The scoring was structured to align with the relevance and desirability of practices. For questions related to less-preferred practices, inverse scoring was used to ensure alignment with the evaluation framework. For instance, Question 2 states “My organization makes decisions by looking at what other organizations are doing”. Since copying what other organizations practice without recognizing the specific context is regarded as an unhelpful practice, a response of “Always” to this question is assigned a low score (Figure 1).

A total score was obtained for each hospital based on the sum across the 19 questions. Based on the total score, sample hospitals were classified into three categories, as given below.

0–54 points (not evidence-based): Decisions are in general based on factors other than the best available evidence.

55–95 points (sometimes compatible with principles of EBM): Decision-making is sometimes compatible with the principles of EBM.

96–133 points (committed and consistent with principles of EBM): Procedures and approach are consistent with the principles of EBM.

### 2.9. Statistical Analysis

Frequencies and means were computed to examine the distribution of evidence-based management practices. We conducted K-means cluster analysis to identify groups of hospitals that were similar in terms of practices, and Pearson correlation tests to assess the strength of associations between key factors driving evidence-based management. All statistical analyses were carried out using Stata software version 14 (StataCorp LLC, College Station, TX, USA).

## 3. Results

A total of 43 eye hospitals from diverse regions participated in the survey, including 20 member eye hospitals of the WAEH and 23 partner hospitals of LAICO. Together, these results provided a comprehensive perspective on evidence-based management practices in eye hospitals worldwide.

### 3.1. Geographic Distribution of Participants

We show the distribution of the number of participant hospitals across the globe in Table 1. The participants represent all the continents except Africa. All 23 LAICO partner hospitals were in Asian countries, contributing to the higher representation of this region in our sample.

### 3.2. Evidence-Based Practices

Based on their scores, participants were categorized into three levels of evidence-based management (EBM) practices. The results, illustrated in Table 1, indicate a mixed level of adoption across regions. Hospitals in Australia, the Middle East, and South America exhibited a strong commitment and consistency in adhering to EBM principles, compared to those in other regions. In contrast, hospitals in Asia, Europe, and North America displayed some practices that were aligned with EBM principles.

### 3.3. Thematic Focuses of Management Practices Across Regions

We computed mean scores based on thematic focus groupings of practices (Figure 2). The analysis revealed that use of external references scored the lowest in all the regions, indicating that most hospitals follow practices in this area that do not align with EBM principles.

Hospitals in North and South America, as well as the Middle East, demonstrated easier access to data and a stronger capacity to understand the quality and implications of the data. These factors contributed to a higher adoption of evidence-based management (EBM) principles in these regions. Conversely, hospitals in Europe, despite having better access to data, scored lower in data-based practices. Interestingly, hospitals in Asia showed stronger adherence to EBM principles compared to Australia and Europe, even though their access to data was comparatively limited.

### 3.4. Associations Among Thematic Categories of Management Practices

We analysed the relationship between evidence-based practices using the thematic focus areas (Table 2). The results of the correlation analysis revealed that evidence-based practices were significantly more prevalent in environments where the ability to access and comprehend data (r = 0.69, *p* < 0.01) and the effective utilization of internal expertise and knowledge (r = 0.48, *p* < 0.01) were observed. Conversely, a low negative correlation was found between the use of external references and guidelines and both evidence-based practices (r = −0.38, *p* < 0.05) and data access and comprehension skills (r = −0.36, *p* < 0.05).

### 3.5. Associations of Thematic Focuses of Management Practice with Hospital Clusters

To identify groups of hospitals that were similar in terms of practices by using their scores on the four thematic focus groups, we conducted a K-means cluster analysis. This yielded three meaningful groups, which were labeled as Cluster A (4 hospitals), Cluster B (14 hospitals), and Cluster C (25 hospitals). In Figure 3, we show the mean scores by thematic areas of management practices.

The three clusters differ systematically on the four thematic areas. Cluster A has the highest external score but lower scores on the other three themes. Cluster C has moderate scores for external references and guidance, but higher scores on the other three themes. Cluster B has lower scores than Cluster C on all four themes. We summarize these differences as follows: Cluster A is a small group of four hospitals that each have a strong external focus but are relatively weak on the use of evidence-based practices, data accessibility, and internal expertise. Cluster B is a group of 14 hospitals that have a moderate scores on evidence-based practices, data accessibility, and internal expertise, but are low on external reference. Finally, cluster C includes 25 hospitals that are strong on evidence-based practices, data accessibility, and internal expertise.

### 3.6. Practices That Support and Undermine Alignment with EBM Principles

Scores for individual questions were analysed to identify specific practices that influenced or undermined the overall scores (Table 3). Practices were ranked in ascending order based on the percentage of hospitals (out of n = 43) that scored high values (5 or 7 on the scale). This analysis highlighted evidence-based management practices that had high versus low prevalence in the sample hospitals.

Practices related to the use of internal and stakeholder expertise (IE) and the ability of the team to access and comprehend data (DA), along with evidence-based practices (EB), scored higher, contributing to overall better performance in these areas. Conversely, practices focused on the use of external references and guidelines (ER) scored lower, emphasizing the need for a more comprehensive understanding of these aspects when implementing external references into management practices.

## 4. Discussion

Evidence-based management (EBM) thrives in organizations where foundational elements—such as leadership commitment [24], a supportive organizational culture, robust infrastructure, data availability, internal expertise [25], and structured processes for integrating evidence—are well-established. These factors work together to ensure the successful implementation of EBM, enhancing strategic planning, operational efficiency, and overall organizational performance. The extent of EBM adoption is largely determined by how effectively these foundational elements are integrated and maintained [6,9]

Our study found that the level of adoption of evidence-based management practices varies significantly among eye hospitals and across regions (Table 1). Organizations that actively engage internal expertise and stakeholders in decision-making processes are more likely to practice evidence-based management (EBM). Moreover, organizations that create an environment conducive to easy access to data and focus on building the team’s capacity to validate data quality tend to exhibit stronger evidence-based practices (Figure 2). Similarly, hospitals that heavily rely on external references tend to exhibit lower adherence to EBM principles (Figure 3). In contrast, hospitals with a skilled team proficient in accessing and interpreting data, coupled with effective utilization of internal expertise, and less dependence on external references, are more likely to practice EBM principles effectively.

Organizational practices that consistently align with evidence-based management (EBM) principles include ensuring access to reliable information, which enables users to effectively engage with data (Table 2) [26]. Additionally, building the capacity of the team to evaluate data quality fosters informed decision-making [6]. Maintaining a discipline of consulting experienced staff and stakeholders fosters collaborative and well-rounded problem-solving. Leveraging internal data to identify challenges and devise actionable solutions is a crucial component of effective decision-making. Additionally, implementing structured systems and processes to learn from internal data and past mistakes cultivates a culture of continuous improvement [27]. Together, these practices form a strong foundation for integrating evidence-based management (EBM) principles into organizational workflows.

Management practices that rely on external references and guidelines—such as adopting innovative approaches, benchmarking against other organizations, making decisions based on external models, or following external consultant recommendations—do not effectively align with evidence-based management (EBM) principles (Table 3). While these practices can be valuable, they must be approached with caution. Innovation is beneficial when it addresses a clearly defined problem; however, decisions made without fully understanding the practical implication of these innovations may lead to inefficiencies. Benchmarking, while insightful, might not yield meaningful results if the organizational context in which the practice was successful is not fully understood [28,29]. Similarly, decisions modelled on other organizations’ strategies may lose relevance due to differences in context or a lack of access to crucial details, such as the underlying rationale and conditions for success.

While numerous studies have identified factors that enable evidence-based management (EBM) practices, our study emphasizes specific enablers such as the capacity of teams to access, assess, and interpret data, as well as the effective utilization of internal resources, including organizational data and the experience and knowledge of senior staff and stakeholders. Conversely, practices based on external references and guidelines may hinder an organization’s ability to access and use internal data, leverage internal expertise, and implement evidence-based practices; collectively, these practices do not align with evidence-based management (EBM) principles. Although EBM significantly contributes to establishing a structured foundation for informed decision-making and continuous improvement, performance outcomes in healthcare organizations are influenced by various factors, including leadership, resource availability, organizational culture, and the external environment. As such, performance variations among organizations cannot be attributed solely to EBM. Nonetheless, the impact of EBM on performance remains a nuanced topic warranting deeper exploration.

Our study included participants from all continents with the exception of Africa, which is a significant strength. However, it did not evaluate performance variations in relation to the adoption levels of EBM, representing a limitation of the study. Also, the data are based on self-reports by organizational leaders, who are expected to have a comprehensive understanding of the current practices of their organization. However, it is possible that the respondents may have provided socially desirable responses, which could introduce overstatement bias. The number of respondents at the regional level limits the depth of the analysis between regions. Additionally, we acknowledge that cultural diversity, given the extensive geographical representation, may have contributed to variations in practices. This also presents an opportunity for future research to explore how local and organizational cultures influence decision-making processes, potentially using, for example, Geert Hofstede’s cultural dimensions framework to gain deeper insights into the cultural factors shaping organizational practice.

## 5. Conclusions

Eye hospitals worldwide collectively demonstrate a strong adoption of practices aligned with evidence-based management (EBM), with variations observed across regions. Most hospitals were categorized in the highest tier of adherence to EBM principles, namely, exhibiting commitment and consistency. These findings highlight that EBM is more effectively practiced when organizations possess the capacity to assess and utilize internal data, leverage the expertise of senior staff and stakeholders, and implement robust systems for continuous improvement. While referencing practices from other organizations can be valuable, caution is necessary to ensure that critical contextual details, rationales, and implementation strategies are understood and appropriately adapted.

To enhance adherence to EBM, hospitals should address practices that undermine these principles and work towards aligning current operations with evidence-based approaches. This alignment fosters informed decision-making and leads to improved management outcomes. Additionally, organizing targeted educational sessions can support hospitals in adopting practices that align with EBM, further strengthening the capacity of these institutions for evidence-based management.

## Figures and Tables

**Figure 1 healthcare-13-00222-f001:**
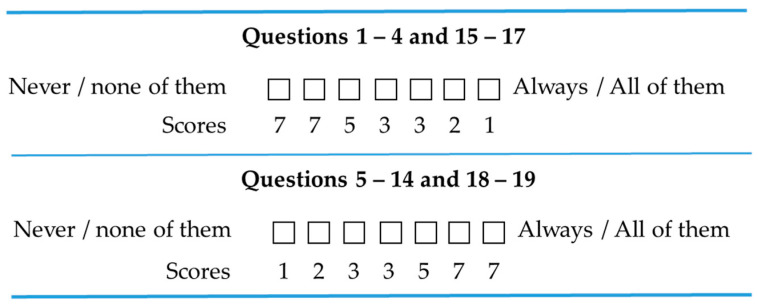
Scoring of responses.

**Figure 2 healthcare-13-00222-f002:**
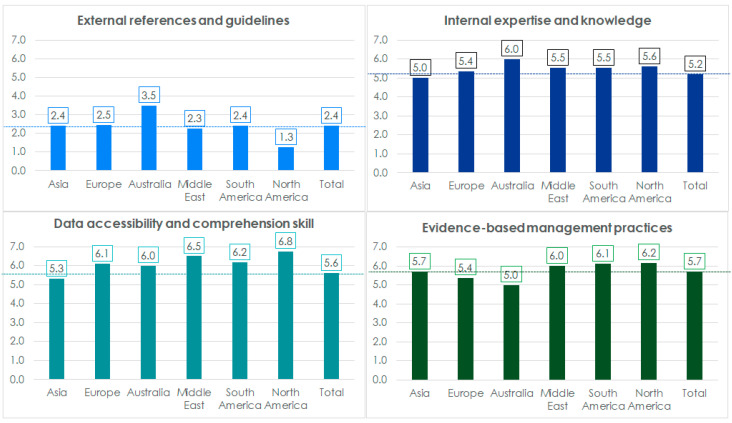
Mean scores, by thematic focus, of management practices across regions.

**Figure 3 healthcare-13-00222-f003:**
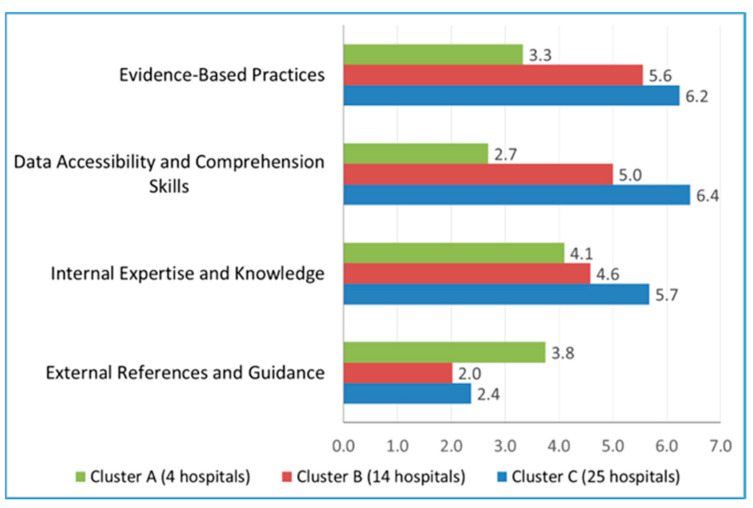
K-means clusters of participant hospitals based on the thematic focus areas of management practices.

**Table 1 healthcare-13-00222-t001:** Categorization of participant hospitals into evidence-based management practice groups based on total score, by region.

Regions n, Row %	Not Evidence-Based	Sometimes Compatible with Principles of EBM	Committed and Consistent with Principles of EBM	Total
Asia	1	17	11	29
	3.5%	58.6%	37.9%	100%
Australia	0	0	1	1
	0.0%	0.0%	100.0%	100%
Europe	0	3	2	5
	0.0%	60.0%	40.0%	100%
Middle East	0	1	2	3
	0.0%	33.3%	66.7%	100%
North America	0	1	1	2
	0.0%	50.0%	50.0%	100%
South America	0	1	2	3
	0.0%	33.3%	66.7%	100%
Total	1	23	19	43
	2.3%	53.5%	44.2%	100%

**Table 2 healthcare-13-00222-t002:** Correlations of mean scores of thematic focuses of management practice (The *p*-values are in parentheses; bold indicates *p* < 0.05.)

Variables Co-Efficient and *p*-Value	Evidence-Based Practices	Use of External References and Guidelines	Use of Internal Expertise	Data Access and Comprehension Skills
Evidence-based practices	1.00	−0.38 (0.01)	0.48 (0.00)	0.69 (0.00)
Use of external references and guidelines	−0.38 **(0.01)**	1.00	−0.23 (0.13)	−0.36 (0.02)
Use of internal expertise	0.48 **(0.00)**	−0.23 (0.13)	1.00	0.37 (0.02)
Data access and comprehension skills	0.69 **(0.00)**	−0.36 **(0.02)**	0.37 **(0.02)**	1.00

**Table 3 healthcare-13-00222-t003:** Practices deviating from or adhering to EBM principles.

Thematic Group	Questions	Scores (Number of Responses Shown in Table)					Group	
		1	2	3	5	7	% Who Scored 1, 2 or 3	% Who Scored 5 or 7
ER	My organization believes it is important to adopt new and innovative practices.	29	11	3	0	0	100%	0%
ER	My organization uses benchmarking to identify best practices used in other organizations.	15	14	11	1	2	93%	7%
ER	My organization makes decisions by looking at what other organizations are doing.	11	5	21	3	3	86%	14%
ER	My organization uses consultants to help us make decisions.	6	8	21	3	5	81%	19%
IE	Managers in my organization tend to believe that the organization is unique and hence the outcome of scientific research is not applicable.	1	6	15	6	15	51%	49%
IE	Managers and senior staff in my organization tend to believe that experience and knowledge gained on the job is the only important source of information when considering how to tackle a problem.	4	7	11	7	14	51%	49%
IE	Internal politics and power struggles influence the way my organization makes decisions about policies and practices.	2	10	7	6	18	44%	56%
EB	We follow a systematic improvement process to address the problems and work on improvements. e.g., Lean six sigma, IHI (Institute of Health Improvement), TPM, TQM model etc.	3	1	12	13	14	37%	63%
DA	Managers in my organisation know how to appraise the trustworthiness of the findings from scientific research.	3	4	7	13	16	33%	67%
EB	Before any (strategic/tactical/important) decision is taken, my organisation consults the scientific literature to verify claims regarding assumed problems or effective solutions.	0	0	10	12	21	23%	77%
EB	Our organisation systematically evaluates the effectiveness of new policies and practices before we introduce.	1	1	8	10	23	23%	77%
EB	Before we implement new policies or practices we obtain a baseline against which subsequent evaluation can be compared.	0	2	7	9	25	21%	79%
DA	Managers and senior staff in my organisation know how to use the Internet to search for scientific evidence to guide their decisions.	1	2	5	12	23	19%	81%
IE	Before any (strategic/operational/routine/tactical) decision is taken, we consult the most important stakeholders (people inside or outside the organisation who may be affected by the decision) to verify claims regarding assumed problems or effective solutions.	0	0	6	5	32	14%	86%
DA	Managers and senior staff in my organisation have access to appropriate information systems that contain People Data (e.g., absenteeism, turnover, job satisfaction), Patients Data (Number of patients served, Number of complaints, Patients’ satisfaction) and overall Business Performance Data (e.g., productivity data, financial indicators)	0	0	6	9	28	14%	86%
DA	Managers and staff in my organisation know how to critically assess the quality of organisational/internal data.	0	1	4	10	28	12%	88%
IE	Before any (strategic/operational/routine/tactical) decision is taken we consult experienced professionals within our organisation to verify claims regarding assumed problems or effective solutions.	0	1	3	7	32	9%	91%
EB	Before any (strategic/operational/routine/tactical) decision is taken, we systematically evaluate internal data to better understand the nature of the problem	0	1	3	5	34	9%	91%
EB	If we make mistakes in our decision-making we try to learn from them.	0	0	2	4	37	5%	95%

Abbreviations: ER—External references and guidelines; EB—Evidence-based practices; IE—Use of internal expertise; DA—Data access and comprehension skills.

## Data Availability

Data are available in the publicly accessible repository. The original raw data are freely available via the following DOI: https://doi.org/10.5281/zenodo.14502143 (accessed on 14 December 2024).

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
