# Peer review of "Evaluating the Adoption of Evidence-Based Management Practices in Eye Hospitals"

_healthcare, 2025, doi:10.3390/healthcare13030222_

Round 1

Reviewer 1 Report

Comments and Suggestions for Authors

Who answered the questions of questionnaire ?

Some questions will require inputs from multiple people

Was questionnaire available in public domain  ?

Was there a common language ?? Or it required translation

Promotional content should be avoided in the academic articles; authors are suggested to decrease the write up on

LAICO and WAEH to minimum.

As per the website  Evidence-Based Management Assessment for Organizations: is not a validated assessment tool, rather a set of statements that can be used to stimulate discussion and think about what evidence-based management might mean to you, your team, and your organization.

Author Response

Dear Reviewer 1,

Respected Reviewer, We sincerely thank you for taking the time to review our manuscript and for providing constructive feedback. Your valuable suggestions have significantly contributed to enhancing the quality and readability of the content. We have carefully addressed all your comments, and a point-by-point response detailing the changes made in the manuscript using track changes, is provided for your review.

Comments-1: Who answered the questions of questionnaire?

Response 1: Thank you for highlighting this important information that we missed. We have now included the following sentence: “We requested the Head of the organization to respond to this survey.” Page 5: section 2.6.1 and 2.6.2.

Comments-2: Some questions will require inputs from multiple people.

Response-2: Thank you for raising this clarification.  The questionnaire was completed by the head of the organization, who is expected to be knowledgeable about and actively managing the current practices in the organization. However, it is possible that the respondent may not have a complete understanding of the ground realities within their organization or may have provided socially desirable responses. We now acknowledge this limitation in the discussion section. Page 12 (before 5. Conclusion)

Comments-3: Was questionnaire available in public domain? 

Response-3: Thank you.  We have now included the phrase ‘a public domain tool’ in section 2.4

Comments-4: Was there a common language?? Or it required translation –

Response-4: Thank you for pointing this out.  We have now added a sentence “We used the same questionnaire in its English version for all respondents. Considering the profile of the respondents, translations were deemed unnecessary”; Page 3: Section 2.3

Comments-5: Promotional content should be avoided in the academic articles; authors are suggested to decrease the write up on LAICO and WAEH to minimum.

Response-5: Thank you for this suggestion.  We have minimised the description of LAICO and WAEH to the sentences shown below – Page-3: Section 2.1

LAICO (https://laico.org), the training and consulting division of the Aravind Eye Care System in India, was founded in 1992 with a mandate to support eye care programs globally.

WAEH (https://www.waeh.org), a global network of independent specialized eye hospitals, promotes collaborative learning and the exchange of best practices.

Comments-6:  As per the website Evidence-Based Management Assessment for Organizations: is not a validated assessment tool, rather a set of statements that can be used to stimulate discussion and think about what evidence-based management might mean to you, your team, and your organization.

Response-6:  Thank you for pointing this out.  We have now added “As noted on the CEBMa website, the tool has not been formally validated. In Page 3, Section 2.3. However, we did validate it. The details of the validation are given in Section 2.5.

Thank you.

Reviewer 2 Report

Comments and Suggestions for Authors

Thank you for the opportunity to review the manuscript. The study of evidence-based management within healthcare institutions represents a critical area of research, offering substantial potential for advancements in healthcare delivery and organisational practices. It is interesting and worth reading to both management and healthcare researchers and practitioners.

The introduction includes major issues and a brief literature review. The cited works, such as E. Lehane et al. [2], T. Shafaghat et al. [6], S. Humphries et al. [7], and A. Janat [9], span a period from 2014 to 2021, reflecting the evolving perspectives within the field.

The empirical research is based on a questionnaire developed by scholars from an institution well-known in the evidence-based management community. The questionnaire has been validated and tailored in the concrete context of the presented manuscript. Section 2.5 provides a detailed thorough account of the research design, including the preparation process, respondent demographics, and data collection methodology. 

The results of the findings are presented in the manuscript in Table 2 and Figure 1, and the raw data is presented as a supplement to ensure the transparancy and reliability of the findings.

However, the authors might wish to consider several comments. The Introduction section could be refined by concluding with an explicit outline of the manuscript’s structure. This addition would assist readers in navigating the study more effectively.

A detailed explanation of how ethical issues were addressed during the research process would reinforce the credibility and reliability of the findings. Even if the research does not involve the patients and does not reveal personal data, the healthcare sector requires additional care regarding the ethics of the research.

The reference list requires careful revision. For example, sources [5], [22], and [28] contain inconsistencies in author names, and the remaining references should be checked for accuracy and completeness.

The analysis of the findings presented in Figure 1 and Table 2 could be expanded. A deeper exploration of the relationships among the four criteria groups and their intercorrelations would provide more nuanced insights, thereby enriching the study’s contribution to the field

The number of respondents is limited, even if justified for this particular research. It is particularly notable given the extensive geographical and cultural diversity of the participating hospitals. Future research could address these limitations by comparing results across regions. Incorporating frameworks such as G. Hofstede’s Cultural Dimensions Theory or similar scholarly models may yield valuable insights into how local and organizational cultures influence decision-making processes.

These recommendations are intended to enhance the clarity and impact of the manuscript. They do not question the quality of the presented research but rather aim to facilitate broader comprehension and application of its findings within the academic community.

Author Response

Dear Reviewer 2,

Respected Reviewer, We sincerely thank you for taking time to review the manuscript and provide your feedback to enrich the manuscript. We have carefully read all your comments and addressed them appropriately in the manuscript using track changes.  We provide the details of the corrections and additions made in the manuscript in the following section.

Comments-1:  The Introduction section could be refined by concluding with an explicit outline of the manuscript’s structure. This addition would assist readers in navigating the study more effectively.

Response-1:  Thank you for this suggestion to assist the readers in navigating the manuscript.  We have now included the following paragraph at the end of section-1 on Page-2

The remainder of this manuscript is organized as follows: The subsequent sections outline the survey administration process, questionnaire validation, response grouping and scoring, geographic distribution of participants, the alignment of current practices with Evidence-Based Management (EBM) principles, and the thematic focus of management practices across regions. Furthermore, these sections examine the relationship between management practices, identify practices that hinder alignment with EBM principles, and conclude with a comprehensive discussion of the findings.

Comments-2: A detailed explanation of how ethical issues were addressed during the research process would reinforce the credibility and reliability of the findings. Even if the research does not involve the patients and does not reveal personal data, the healthcare sector requires additional care regarding the ethics of the research.

Response-2: Thank you for this suggestion.  We have now included a sentence “Respondents' identification details, including both institutional and individual names, are securely anonymized and stored to ensure confidentiality” in Page-3: Section 2.4.

Comments-3: The reference list requires careful revision. For example, sources [5], [22], and [28] contain inconsistencies in author names, and the remaining references should be checked for accuracy and completeness.

Response-3: Thank you for highlighting this error.  They are now corrected as follows.

[5]         R. Guo, S. D. Berkshire, L. Fulton, and P. Hermanson, “Using an Evidence-Based Management Approach in Healthcare Administration Decision-Making,” Acad. Manag. Proc., vol. 2017, no. 1, p. 10240, 2017, doi: 10.5465/ambpp.2017.10240abstract.

[22]       A. Goncharuk, N. O. Lazareva, and I. A. M. Alsharf, “Benchmarking as a performance management method,” Polish J. Manag. Stud., vol. 11, No2, no. July 2015, pp. 27–36, Jul. 2016.

[28]       J. Denrell, “Selection bias and the perils of benchmarking.,” Harv. Bus. Rev., vol. 83, no. 4, pp. 114-119,134, Apr. 2005.

Comments-4:  The analysis of the findings presented in Figure 1 and Table 2 could be expanded. A deeper exploration of the relationships among the four criteria groups and their intercorrelations would provide more nuanced insights, thereby enriching the study’s contribution to the field

Response-4:  Thank you for this important suggestion. We have made table-2 complete to show the intercorrelations.  As suggested for a deeper exploration of the relationship among the four criteria group, we conducted K-means cluster analysis and presented it in Figure-2. Appropriate interpretations are included in the results and discussion sections.

Comments-5: The number of respondents is limited, even if justified for this particular research. It is particularly notable given the extensive geographical and cultural diversity of the participating hospitals. Future research could address these limitations by comparing results across regions. Incorporating frameworks such as G. Hofstede’s Cultural Dimensions Theory or similar scholarly models may yield valuable insights into how local and organizational cultures influence decision-making processes.

Response-5: Thank you for your valuable suggestions and the opportunity to explore this further from different perspectives. We have now included the following sentence in the limitations section: 'We recognize that cultural diversity, given the extensive geographical representation, may have contributed to variations in practices. This also presents an opportunity for future research to explore how local and organizational cultures influence decision-making processes, shaping what are considered evidence-based management practices and how they are utilized, using Geert Hofstede's cultural dimensions framework (Page-11: prior to Section 5. Conclusion)

These recommendations are intended to enhance the clarity and impact of the manuscript. They do not question the quality of the presented research but rather aim to facilitate broader comprehension and application of its findings within the academic community.

We deeply appreciate your thoughtful comments on our manuscript and sincerely thank you for taking the extra effort to highlight further research opportunities in this area. 

Thank you.

Reviewer 3 Report

Comments and Suggestions for Authors

1. What is the confidence interval of the project. The confidence interval may affect the sampling rate of the research.

2. What is the inclusion and exclusion criteria of the eye hospital. Is the 95 eye hospital covered all the continents in the world (except Africa)? 

3. How the scoring is assigned in the survey, for example, line 220. How the authors assign 7 to both metrics (at the left). 

4. Are question validity performed prior to the survey? What methodology is employed.

Author Response

Dear Reviewer 3,

Respected Reviewer, We sincerely thank you for taking time to review our manuscript and sharing the valuable feedback.  Your comments are very valuable in addressing some of the lacunae in the manuscript, which are important for the readers to know.  We have addressed your comments in the manuscript appropriately using track changes and also given explanation for your question in the point by point

Comments-1. What is the confidence interval of the project. The confidence interval may affect the sampling rate of the research.

Response-1: Thank you for raising this question. Since we approached all members of WAEH (i.e., conducted a census) and used convenience sampling to select active partner hospitals of LAICO (i.e., conducted non-random sampling), we decided it was not appropriate to compute confidence intervals which would require an assumption of sampling error. Page #3, Section 2.2.

Comments-2. What is the inclusion and exclusion criteria of the eye hospital? Is the 95 eye hospital covered all the continents in the world (except Africa)?  

Response-2: Thank you for pointing out this missing detail in the manuscript. We expanded as “We selected all 61 member hospitals of WAEH and selected 33 partner hospitals of LAICO’s active collaborative network. The selection criteria focused on eye hospitals that had been actively engaged during the three years preceding the survey”. Page-3 Section 2.2

Comments-3. How the scoring is assigned in the survey, for example, line 220. How the authors assign 7 to both metrics (at the left). 

Response-3: Thank you. We adhered to the scoring methodology specified in the questionnaire tool, (Page-6, Section 2.8) without any modifications. While we do not have details on how the scoring was determined, we assumed that the experts had assigned equal weightage to the top two categories, as there may be a similar effect between 'Always' and 'Nearly Always.' We have given the reference [23] for the further details on the scoring methodology.

Comments-4. Are question validity performed prior to the survey? What methodology is employed.

Response-4:  Thank you for raising this clarification. As it was explicitly stated in the questionnaire that 'it is not a validated tool,' we conducted our own validation process, focusing on readability, comprehensibility, and the order of the questions before implementing it with survey participants. The questionnaire was initially tested among senior managers at Aravind Eye Hospital, Madurai. Responses were analysed to assess internal consistency and reliability, yielding a Cronbach’s alpha of 0.84, which indicates strong reliability. These details are provided on Page #5, under Section 2.5.

Thank you.

Round 2

Reviewer 3 Report

Comments and Suggestions for Authors

Comments are addressed